# A Holistic Analysis of Individual Brain Activity Revealed the Relationship of Brain Areal Activity with the Entire Brain’s Activity

**DOI:** 10.3390/brainsci13010006

**Published:** 2022-12-20

**Authors:** Jie Huang

**Affiliations:** Department of Radiology, Michigan State University, East Lansing, MI 48824, USA; huangj@msu.edu; Tel.: +1-517-884-3246

**Keywords:** BOLD-fMRI, functional area of unitary pooled activity, FAUPA, brain areal activity, entire brain activity

## Abstract

The relationship between brain areal activity and the entire brain’s activity is unknown, and understanding this relationship is imperative for understanding the neural mechanisms of human brain function at systems level. The complex activity of human brains varies from area to area and from time to time across the whole brain. BOLD-fMRI measures this spatiotemporal activity at a large-scale systems level. The BOLD time signal of an area reflects a collective neuronal activity of over one million neurons under that area, and the temporal correlation of this time signal with that of every point in the brain yields a full spatial map that characterizes the entire brain’s functional co-activity (FC) relative to that area’s activity. Here we show a quantitative relationship between brain areal activity and the activity of the entire brain. The temporal correlation coefficient *r* of the signal time courses of two areas quantifies the degree of co-activity between the two areas, and the spatial correlation coefficient *R* of their corresponding two FC maps quantifies the co-activity between these two maps. We found that a modified sigmoid function quantified this *R* with *r*, i.e., $R\left(r\right)=\frac{{\left(1+r\right)}^{a}-{\left(1-r\right)}^{a}}{{\left(1+r\right)}^{a}+{\left(1-r\right)}^{a}}$, revealing a relationship between the activity of brain areas and that of the entire brain. The parameter a in this equation was found to be associated with the mean degree of the temporal co-activity among all brain areas, and its value was brain functional state dependent too. Our study demonstrated a novel approach for analyzing fMRI data to holistically characterize the entire brain’s activity quantitatively for any brain functional state in individual humans.

## 1. Introduction

The non-invasive blood-oxygenation-level-dependent (BOLD) functional magnetic resonance imaging (fMRI) technique has evolved as a major neuroimaging tool for investigating in vivo human brain functional organization of neural activity at a large-scale systems level [1,2]. The BOLD signal measures the underlying neural activity indirectly through the neurovascular coupling mechanism that links transient neural activity to the subsequent change in cerebral blood flow [3,4,5]. In a typical task-fMRI study with a spatial resolution of 3 × 3 × 3 mm^3^ and a temporal resolution of 2 s, a task-activated voxel may contain over one million neurons and its corresponding BOLD signal change measures the activity-induced change from a pooled activity of these million neurons [6]. Numerous fMRI studies reported in the literature have demonstrated its effectiveness and reliability in investigating human brain activity at a group level (i.e., the commonality across the subjects within a group) and the effects of brain disorders on brain activity. Although investigating this commonality is important in understanding human brain functional organization, studying individual brain functioning may be imperative for understanding the neural bases responsible for individual behavioral and clinical traits. Recently, the BOLD-fMRI technique has been utilized to study individual brain functional organization and its relationship to behavior, demonstrating substantial promise of a person-specific neuroimaging approach in investigating individual brain functioning [7,8,9,10,11,12,13,14,15,16,17,18,19,20,21]. Here, we present a novel data-driven method to analyze fMRI data objectively and to holistically characterize the entire brain’s activity quantitatively for any brain functional state in individual humans.

Brain activity is complex and dynamic; it varies from area to area and from time to time across the whole brain. BOLD-fMRI measures this spatiotemporal activity at a large-scale systems level. We recently conceived the concept of human brain functional areas of unitary pooled activity (FAUPAs) and developed a method to identify FAUPAs with fMRI [22]. A FAUPA is defined as an area in which the temporal variation of the activity is the same across the entire area, i.e., the neuronal mass activity is a unitary dynamic activity across the entire area. This dynamically unitary activity implies a perfect temporal correlation (TC) everywhere within the FAUPA for the neuronal mass activity-induced BOLD response, i.e., the corresponding TC *r* is 1 for the BOLD responses of any two locations within the FAUPA. FAUPAs were identified for both resting state (rs) and task-fMRI, and their determination was objective and automatic with no a priori knowledge. The majority of the identified FAUPAs were located within the cerebral cortex and, for those within the cerebral cortex, they were located mainly within the gray matter, indicating an association of their dynamic signal changes with the underlying neuronal activities.

The whole brain activity is reflected in the collective activities of all voxels within the entire brain. For each identified FAUPA, the TC *r* of that FAUPA’s signal time course with that of every voxel within the brain yields a spatial functional co-activity (FC) map across the entire brain. This FC map characterizes the spatial distribution of the entire brain’s co-activity relative to that of the FAUPA, providing a measure to quantify whole brain activity to some degree. For a given brain state (i.e., resting or task), the TC *r* of signal time course between two FAUPAs quantifies the degree of co-activity between them. Similarly, for their corresponding two FC maps, the spatial correlation (SC) *R* of these two maps across all voxels within the brain quantifies the co-activity between the two maps. As the FAUPAs’ activities are parts of the entire brain’s activity, the TC *r* and SC *R* may be related to each other. Examining the TC *r* of pairwise combinations of all FAUPAs with the SC *R* of pairwise combinations of their corresponding FC maps may reveal this relationship. In the present study, we analyzed the same rs- and task-fMRI data reported in our previous study [22], and revealed a quantitative relationship between the TC *r* and SC *R*.

## 2. Materials and Methods

This is a follow-up study of our previous three studies [22,23,24]. This study analyzed the same fMRI data. Accordingly, it used the same subjects, same image acquisition, similar image preprocessing procedures, and same algorithms for FAUPA determination. For more information, refer to our previous study [22].

Participants: nine healthy subjects (5 male and 4 female, ages 21–55 years old) participated in the study.

Image acquisition: functional brain images were acquired on a GE 3.0 T clinical scanner with an 8-channel head coil using a gradient echo Echo-Planar-Imaging pulse sequence (TE/TR = 28/2500 ms, flip angle 80°, FOV 224 mm, matrix 64×64, slice thickness 3.5 mm, and spacing 0.0 mm). Thirty-eight axial slices to cover the whole brain were scanned, and the first three volume images were discarded. Each subject undertook a 12 min rs-fMRI scan and a 12 min task-fMRI scan. For the rs-fMRI scan, the subjects were instructed to close their eyes and try not to think of anything but remain awake during the whole scan. The task paradigm consisted of a total of 24 task trials with 3 different tasks of reading words, tapping fingers, and viewing patterns. Each task trial comprised a 6 s task period followed by a 24 s resting period. (For further details, refer to our previous study [22].)

Image preprocessing: image preprocessing of the functional images was performed using AFNI (analysis of functional neuro images) software [22,25]. It included removing spikes, slice-timing correction, motion correction, spatial filtering with a Gaussian kernel with a full-width-half-maximum of 4.0 mm, computing the mean volume image, bandpassing the signal intensity time courses to the range of 0.009–0.08 Hz, and computing the relative signal change (%) of the bandpassed signal intensity time courses. After these preprocessing steps, further image analysis was carried out using in-house developed Matlab-based software algorithms.

FAUPA determination: a statistical model and Matlab-based software algorithms were developed and tested to determine FAUPAs, and FAUPAs were identified and reported for both the rs-fMRI and task-fMRI [22,26]. The FAUPA determination involved the iterative aggregation of voxels dependent upon their intercorrelation, and the algorithms were described in detail in our previous study [22]. The determination consists of two major procedures. (1) Using a first statistical criterion the algorithm first identifies a stable region of interest (ROI) in which the signal time courses of all voxels show a similar temporal behavior. (2) Using a second statistical criterion it determines whether this stable ROI satisfies the condition of being a FAUPA by comparing the temporal behavior of signal time course of the voxels within the FAUPA with those bordering the FAUPA.

Computations of *r*, *R*, the distribution density of *r*, and its mean *r*: for each identified FAUPA, we first computed the TC coefficient of that FAUPA’s signal time course $f\left(t_{i}\right)$ with every voxel across the whole brain to yield its corresponding FC map $F\left(v_{j}\right)$, i.e.,
(1)$$F\left(v_{j}\right)=\frac{\sum_{t_{i}=1}^{T}\left[v_{j}\left(t_{i}\right)-\bar{v_{j}}\right]\left[f\left(t_{i}\right)-\bar{f}\right]}{\sqrt{\left\{\sum_{t_{i}=1}^{T}{\left[v_{j}\left(t_{i}\right)-\bar{v_{j}}\right]}^{2}\right\}\left\{\sum_{t_{i}=1}^{T}{\left[f\left(t_{i}\right)-\bar{f}\right]}^{2}\right\}}}\;\;\;\left(v_{j}=1,\;2,\;\cdots \cdots V\right)$$
where $\bar{f}=\frac{1}{T}\sum_{t_{i}=1}^{T}f\left(t_{i}\right)$ is the FAUPA’s mean signal, $\bar{v_{j}}=\frac{1}{T}\sum_{t_{i}=1}^{T}v_{j}\left(t_{i}\right)$ is the mean signal of *j*th voxel, and T is the total number of time points. (A FAUPA’s signal time course is the mean signal time course averaged over all voxels within that FAUPA.) Then, for all pairwise combinations of all identified FAUPAs in each brain state of each subject, we computed the TC *r* of each paired FAUPAs $f_{1}\left(t_{i}\right)$ and $f_{2}\left(t_{i}\right)$,
(2)$$r=\frac{\sum_{t_{i}=1}^{T}\left[f_{1}\left(t_{i}\right)-\bar{f_{1}}\right]\left[f_{2}\left(t_{i}\right)-\bar{f_{2}}\right]}{\sqrt{\left\{\sum_{t_{i}=1}^{T}{\left[f_{1}\left(t_{i}\right)-\bar{f_{1}}\right]}^{2}\right\}\left\{\sum_{t_{i}=1}^{T}{\left[f_{2}\left(t_{i}\right)-\bar{f_{2}}\right]}^{2}\right\}}}$$
and the SC *R* of their corresponding FC maps $F_{1}\left(v_{j}\right)$ and $F_{2}\left(v_{j}\right),$
(3)$$R=\frac{\sum_{v_{j}=1}^{V}\left[F_{1}\left(v_{j}\right)-{\bar{F}}_{1}\right]\left[F_{2}\left(v_{j}\right)-{\bar{F}}_{2}\right]}{\sqrt{\left\{\sum_{v_{j}=1}^{V}{\left[F_{1}\left(v_{j}\right)-{\bar{F}}_{1}\right]}^{2}\right\}\left\{\sum_{v_{j}=1}^{V}{\left[F_{2}\left(v_{j}\right)-{\bar{F}}_{2}\right]}^{2}\right\}}}$$
where $\bar{F}=\frac{1}{V}\sum_{v_{j}=1}^{V}F\left(v_{j}\right)$ and *V* is the total number of voxels within the whole brain. To determine the *r*-*R* curve, we first divided the whole *r* range [−1, 1] to 200 equal intervals with interval size 0.01, and then computed the mean value of *r* for each interval and its corresponding mean *R* value for that interval, resulted in the *r*-*R* curve for each brain state of each subject. To determine the distribution density of *r* across the range [−1, 1], we first counted the total number of paired FAUPAs (*NPF*) in each *r* interval, i.e., *NPF*(*r*), and then calculated its percentage (%) relative to the total number of all pairwise combinations of the FAUPAs (*NAPCF*), i.e.,
(4)$$D\left(r\right)=\frac{NPF\left(r\right)\times 100}{NAPCF}\;\;\;\;\left(\%\right)$$
where *D*(*r*) denotes the density of *r* in the given *r* interval. For each distribution density of *r*, its *mean r* was computed as
(5)$$mean\;r=\frac{\sum_{r=-1}^{1}D\left(r\right)\times r}{N}\;\;\;\left(\%\right)\;\;\;\;and\;N\leq 200$$
where *N* is the total number of nonzero *D*(*r*), i.e., *N* = 200 if *D*(*r*) > 0 for all the 200 *r* intervals.

Let *R*(*r*) denotes the SC *R* as a function of the TC *r* between two FAUPAs $f_{1}\left(t_{i}\right)$ and $f_{2}\left(t_{i}\right)$. This function has three theorems.

**Theorem** **1.**
*R(1) = 1 and R(−1) = −1.*


**Proof of Theorem** **1.**When $f_{1}\left(t_{i}\right)=f_{2}\left(t_{i}\right)$, we have ${\bar{f}}_{1}={\bar{f}}_{2}$ and *r* = 1 from Equation (2). From Equation (1) we have $F_{1}\left(v_{j}\right)=F_{2}\left(v_{j}\right)$. Thus, ${\bar{F}}_{1}={\bar{F}}_{2}$ and *R* = 1 from Equation (3). Similarly, when $f_{1}\left(t_{i}\right)=-f_{2}\left(t_{i}\right)$, we have ${\bar{f}}_{1}=-{\bar{f}}_{2}$ and *r* = −1 from Equation (2). Accordingly, $F_{1}\left(v_{j}\right)=-F_{2}\left(v_{j}\right)$, ${\bar{F}}_{1}=-{\bar{F}}_{2}$ and *R* = −1. □

**Theorem** **2.**
*R(−r) = −R(r).*


**Proof of Theorem** **2.**Consider $\overset{´}{f_{2}}\left(t_{i}\right)=-f_{2}\left(t_{i}\right)$; we have ${\bar{\overset{´}{f}}}_{2}=-{\bar{f}}_{2}$ and $\overset{´}{r}=-r$ from Equation (2). From Equation (1) we have ${\overset{´}{F}}_{2}\left(v_{j}\right)=-F_{2}\left(v_{j}\right)$. Thus, ${\bar{\overset{´}{F}}}_{2}=-{\bar{F}}_{2}$ and $\overset{´}{R}=-R$ from Equation (3). Accordingly, we have *R*(−*r*) = −*R*(*r*). □

**Theorem** **3.**
*R(0) = 0.*


**Proof of Theorem** **3.**As *R*(−*r*) = −*R*(*r*), we have *R*(*r*) + *R*(−*r*) = 0, which yields *R*(0) = 0 because *R*(0) = *R*(−0). □

Here we introduce the function:(6)$$R\left(r\right)=\frac{{\left(1+r\right)}^{a}-{\left(1-r\right)}^{a}}{{\left(1+r\right)}^{a}+{\left(1-r\right)}^{a}}$$
a modified sigmoid function with a range of values from −1 to 1 for both *r* and *R*. Equation (6) satisfies all above three theorems. a is a to be quantified parameter. To quantify the value of a for each brain state of each subject, we defined $Q=\sum_{r}{\left[R\left(r\right)-R_{r}\right]}^{2}$, where $R_{r}$ denotes the SC *R* for the given *r*. Q quantifies the total deviation of $R_{r}$ from $R\left(r\right)$ over all *r* values. Minimizing Q yielded the best fitted value of a for each brain state of each subject.

## 3. Results

### The Relationship between the TC r and SC R

For each brain state (resting vs. task) of each subject, we first computed the full spatial FC map across the entire brain for each identified FAUPA by computing the TC *r* of that FAUPA’s signal time course with that of every voxel within the brain. Then, for all pairwise combinations of these FAUPAs, we computed the TC *r* for each paired FAUPAs and the SC *R* for their corresponding paired FC maps, resulting in an *r*-*R* curve for each brain state of each subject (Figure 1). This *r*-*R* curve showed the same characteristics of sigmoid curves (i.e., a stretched S along the horizontal direction) for both brain states and all subjects, revealing a relationship between brain areal activity and the entire brain’s activity. When two FAUPAs were perfectly positively or negatively correlated (i.e., *r* = 1 or −1), their corresponding FC maps were also perfectly positively or negatively correlated (i.e., *R* = 1 or −1) as shown in each *r*-*R* curve in Figure 1. For strongly correlated FAUPAs, regardless of whether they were positively or negatively correlated, the SC of their corresponding paired FC maps were stronger as shown by these *r*-*R* curves, i.e., the absolute value of *R* is larger than that of *r* (except *r* = 1 or −1), showing the characteristics of the sigmoid curve.

The best fitted *r*-*R* curve using Equation (6) was also obtained for each brain state of each subject, showing a qualitatively good match between the measured *r*-*R* curve and the fitted *r*-*R* curve for every brain state of every subject (Figure 1). Table 1 tabulated the quantified value of a in Equation (6) for each brain state of each subject. To assess the goodness of this fit, we compared the distribution of the standard deviation (SD) of $R_{r}$ with that of the absolute value of the difference between $R_{r}$ and *R*(*r*) for each brain state of each subject and their group mean as well (Figure 2). The difference between $R_{r}$ and *R*(*r*) showed a substantially smaller magnitude of the distribution for both brain states and every subject. For each brain state of each subject, we further computed the mean value of the distribution averaged over all 200 *r* values for both distributions. The difference between $R_{r}$ and *R*(*r*) showed a significantly smaller mean value in comparison to that of the SD of $R_{r}$ for each brain state of each subject and their group mean as well (Table 2).

Although each obtained *r*-*R* curve showed the same characteristics of the sigmoid curve for both brain states and all subjects, the best fitted value of the parameter a in Equation (6) varied substantially from state to state and from subject to subject (Table 1), suggesting a potentially varied brain activity from state to state and from subject to subject. These varied values of a in Equation (6) might reflect those varied activities and, accordingly, were potentially associated with the activities. To investigate these potential associations, for each brain state of each subject we computed the distribution density of the TC *r* of all pairwise combinations of the identified FAUPAs, i.e., *D*(*r*) in Equation (4). This distribution density resulted in a holistic characterization of each brain state for each individual subject (Figure 3). This distributed *r* varied substantially from state to state and from subject to subject, characterizing an individualized brain activity that differed remarkably from subject to subject even for the same functional state. This highly varied *r* distribution should reflect different underlying neural processes because it quantified the correlation of neural activity among all these FAUPAs across the whole brain. It might be responsible for the subtle differences in those *r*-*R* curves in Figure 1. We hypothesized that this *r* distribution determined the parameter value of a in Equation (6). To test this hypothesis, we computed the mean *r* of the *r* distribution averaged over all *r* values (Equation (5)) for each brain state of each subject (Table 1). The parameter a was significantly positively associated with the mean *r* (max *p* < 0.01), showing that the higher the mean degree of the temporal co-activity among all the FAUPAs, the larger the value of a (Figure 3).

## 4. Discussion and Conclusions

Equation (6) quantifies the relationship between the TC *r* of two FAUPAs and the SC *R* of their corresponding two FC maps with the parameter a associated with the mean *r* of the *r* distribution density of pairwise combinations of all FAUPAs. Brain activity varied substantially from state to state and from subject to subject as demonstrated in the remarkably different *r* distribution density from state to state and from subject to subject (Figure 3). As Equation (6) was valid for both functional states (resting vs. task) and every subject (Figure 1), it showed the generality of Equation (6) to brain state and individual subject, revealing a quantitative relationship between the activity of brain areas and that of the entire brain.

The remarkably different *r* distribution from state to state and from subject to subject should reflect individual brain functioning (Figure 3) and might account for individual variability in behavior. It offers a quantitative measure to holistically characterize the entire brain’s activity. As this method of computing the *r* distribution was fully data-driven and consistent for both resting and task states, i.e., the method was independent of the spontaneous and task-evoked brain activity, it offers a novel approach for analyzing fMRI data to investigate each individualized brain activity for any brain functional state in individual humans.

Regarding the person-specific neuroimaging approach for studying individual brain functioning and its relationship to personal traits, it may be worth comparing the presented method with those approaches reported in the literature. First, most fMRI studies analyze data in a standard template space for group analysis, aiming to identify regions of common activation or common functional networks across participants. Such an approach is effective and reliable in identifying the commonality of brain functional organization across a group but may ignore important differences across the participants that might be responsible for individual traits. Individual brains may differ in size and shape, functional areas may vary in anatomical location across individuals, and abnormal brain structure may be associated with neurological disorders [27,28,29,30,31]. Second, although person-specific approaches are effective and reliable in identifying individuals from a group, the frameworks used in these studies are subjective rather than objective as they are defined in a standard template space and then applied to individuals. One common feature in these approaches is that they use either a functional brain atlas consisting of 268 nodes covering the whole brain or group-based parcellations and/or functional networks defined in a standard template space as frameworks to work on [7,17,18,19,20,21]. In contrast, our presented method is objective and fully data-driven with no a priori knowledge, and the analysis is conducted in the original MRI space rather than a standard template space for each individual participant, which might be crucial for applying the BOLD-fMRI technique to daily clinical practice.

Identifying brain functional networks is essential for understanding the functioning of brain systems. Equation (6) quantifies the relationship of the entire brain’s activity with areal activity across the whole brain. The same FC maps reflect the same temporal activity within each of their corresponding FAUPAs, and therefore constitute a functional network with each FAUPA being an architectural unit of the network. Brain functional networks may be identified by identifying all paired FAUPAs with highly correlated FC maps, and then clustering these FAUPAs with the same FC maps to identifying each network. With threshold SC *R* > 0.95, the total number of the paired FAUPAs varied substantially from state to state and from subject to subject (Figure 4), reflecting those dramatically different *r* distributions (Figure 3) and suggesting a possible dependence of the identified functional networks on both brain state and subject. The identification of these individualized functional networks may provide insight into individual brain functioning and personal traits.

Equation (6) further quantifies the strength of interactions between different networks. The definition of FAUPA implies its role of functional unit because the temporal variation of the underlying neuronal mass activity is the same across the entire area within each FAUPA. All FAUPAs that produce the same FC maps constitute the architectural units of a functional network, and their BOLD time signals characterize the dynamic neural activity of that network. For two identified networks, the temporal correlation of two FAUPAs, each from one network, quantifies the degree of the neural co-activity between the two FAUPAs, and the overall correlations of all FAUPAs within one network with those in the other network may characterize the interaction between these two networks. Networks with strong interactions should manifest in relatively large SC *R* values between their corresponding FC maps, but small *R* values for weak interactions, regardless of whether they interact positively or negatively with each other. Two networks with negative SC *R* values interact negatively with each other, i.e., the neural activity increases in one network but decreases in the other one simultaneously, and vice versa. The default mode network is such a well-recognized network [32]. Identifying functional networks and quantifying their interactions may provide a comprehensive analysis of the entire brain’s activity for understanding the functioning of brain systems. As the determination of FAUPAs is objective and automatic with no a priori knowledge, the presented FAUPA method may enable us to objectively identify all functional networks and holistically characterize the entire brain’s activity quantitatively at a large-scale systems level for any brain functional state in individual humans.

## Figures and Tables

**Figure 1 brainsci-13-00006-f001:**
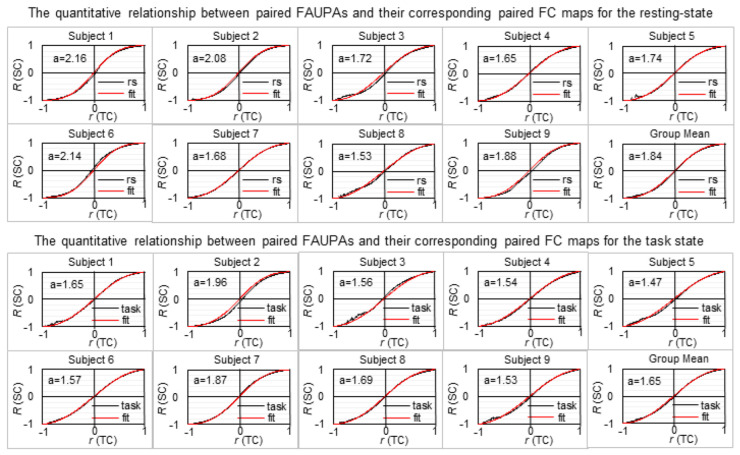
Illustration of the quantitative relationship between brain areal activity and the entire brain’s activity for each brain state of each subject and their group mean as well. The black curve in each plot represents the measured *r*-*R* curve and its corresponding red curve represents the best fitted *r*-*R* curve using Equation (6) for that brain state of that subject. TC: temporal correlation; SC: spatial correlation; rs: resting state.

**Figure 2 brainsci-13-00006-f002:**
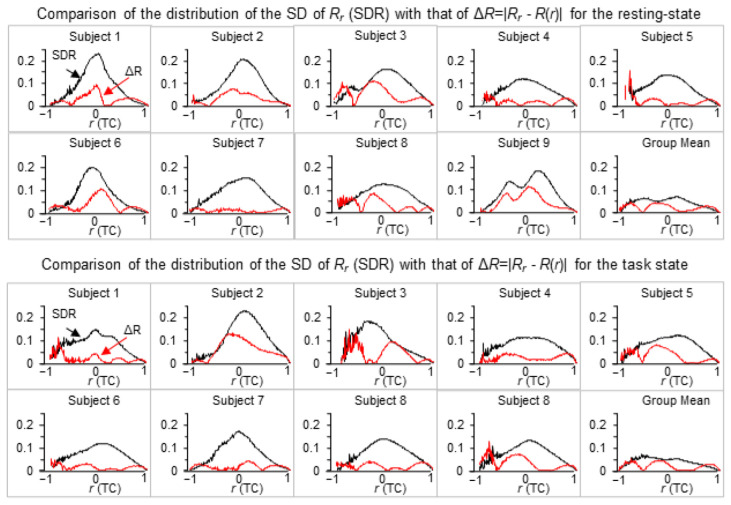
Comparison of the distribution of the standard deviation (SD) of $R_{r}$ as a function of *r* with that of the absolute value of the difference between $R_{r}$ and *R*(*r*), i.e., Δ*R* = |$R_{r}-$
*R*(*r*)|, for each brain state of each subject and the group mean as well. TC: temporal correlation.

**Figure 3 brainsci-13-00006-f003:**
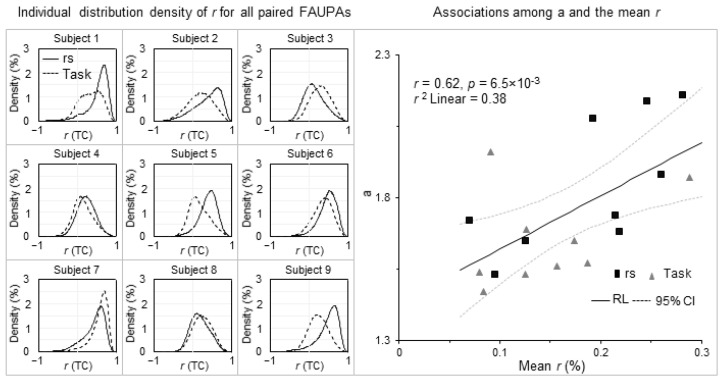
The distribution density of the temporal correlation (TC) *r* of paired FAUPAs for each brain state of each subject (columns 1–3). For a given *r* value, the magnitude of each distribution represented the density that was computed as the percentage (%) of the total number of paired FAUPAs having the same *r* value relative to the total number of all pairwise combinations of the FAUPAs for that state and that subject (Equation (4)). The density-weighted mean of each *r* distribution (Equation (5)) yielded a mean *r* that quantified that distribution to a certain degree: the larger the mean *r*, the stronger the mean co-activity among all the FAUPAs. Note that, for each subject, the difference between the two mean *r* values of the two brain states (Table 1) clearly reflected the difference between their corresponding *r* distributions, and this observation was consistent across all subjects. The right plot illustrates the association of a in Equation (6) with the mean *r*.

**Figure 4 brainsci-13-00006-f004:**
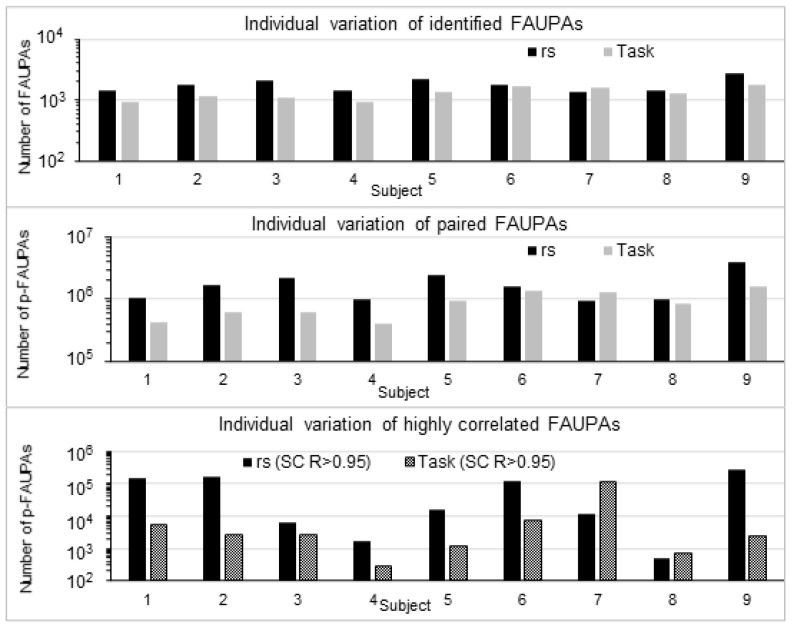
Illustration of individual variation of the identified FAUPAs and highly correlated FAUPAs for both resting and task states of each subject. Top panel: the total number of FAUPAs ranged from 903 to 1784 with mean ± standard deviation (SD) = 1309 ± 320 for the task state and from 1373 to 2804 with mean ± SD = 1818 ± 480 for the resting state (RS), respectively. Middle panel: the total number of all pairwise combinations of N FAUPAs is equal to N × (N − 1)/2. p-FAUPAs: paired FAUPAs. Bottom panel: individual variation of the total number of paired FAUPAs with the threshold SC *R* > 0.95. This number ranged from 288 to 116,230 for the task state and from 520 to 280,521 for the resting state, respectively.

**Table 1 brainsci-13-00006-t001:** The quantified value of a in Equation (6) and the computed value of the mean *r* in Equation (5) for each brain state of each subject. rs: resting-state; SD: standard deviation.

Subject	A		Mean *r*	
	rs	Task	rs	Task
1	2.16	1.65	0.28	0.17
2	2.08	1.96	0.19	0.09
3	1.72	1.56	0.07	0.16
4	1.65	1.54	0.13	0.08
5	1.74	1.47	0.21	0.08
6	2.14	1.57	0.25	0.19
7	1.68	1.87	0.22	0.29
8	1.53	1.69	0.09	0.13
9	1.88	1.53	0.26	0.13
Mean ± SD	1.84 ± 0.23	1.65 ± 0.17	0.19 ± 0.08	0.15 ± 0.07
*t*-test	*p* = 0.06		*p* = 0.23	

**Table 2 brainsci-13-00006-t002:** The mean value of the distribution (Figure 2) averaged over all 200 *r* values for the standard deviation (SD) of $R_{r}$ and the absolute value of the difference between $R_{r}$ and *R*(*r*) (i.e., Δ*R* = |$R_{r}-$
*R*(*r*)|), respectively, for the resting and task states. Note that the last row shows these mean values computed with their corresponding group mean distributions in Figure 2 (the last plot in the second and fourth panels).

Subject	Resting State			Task		
	Mean SD ± SD	Mean Δ ± SD	*p* (*t*-Test)	Mean SD ± SD	Mean Δ ± SD	*p* (*t*-Test)
1	0.088 ± 0.073	0.029 ± 0.022	6.3 × 10^−24^	0.082 ± 0.041	0.022 ± 0.019	3.0 × 10^−56^
2	0.093 ± 0.067	0.035 ± 0.022	7.1 × 10^−26^	0.108 ± 0.075	0.062 ± 0.041	3.6 × 10^−13^
3	0.093 ± 0.048	0.053 ± 0.032	4.6 × 10^−20^	0.095 ± 0.058	0.046 ± 0.035	6.9 × 10^−21^
4	0.073 ± 0.037	0.022 ± 0.012	1.5 × 10^−57^	0.081 ± 0.031	0.025 ± 0.012	3.5 × 10^−72^
5	0.083 ± 0.041	0.021 ± 0.023	1.4 × 10^−56^	0.081 ± 0.035	0.036 ± 0.026	9.8 × 10^−37^
6	0.078 ± 0.066	0.037 ± 0.028	8.1 × 10^−15^	0.073 ± 0.033	0.015 ± 0.010	1.0 × 10^−71^
7	0.086 ± 0.049	0.011 ± 0.006	6.5 × 10^−67^	0.077 ± 0.055	0.018 ± 0.011	1.1 × 10^−39^
8	0.081 ± 0.036	0.036 ± 0.024	2.6 × 10^−37^	0.050 ± 0.042	0.021 ± 0.012	5.4 × 10^−56^
9	0.097 ± 0.055	0.049 ± 0.034	3.6 × 10^−21^	0.077 ± 0.038	0.035 ± 0.027	6.3 × 10^−30^
Group Mean	0.046 ± 0.019	0.025 ± 0.012	1.3 × 10^−33^	0.041 ± 0.017	0.022 ± 0.015	1.0 × 10^−26^

## Data Availability

Both the original and processed fMRI images plus final research data related to this publication will be available to share upon request with a legitimate reason such as to validate the reported findings or to conduct a new analysis.

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
