# Peer review of "A Holistic Analysis of Individual Brain Activity Revealed the Relationship of Brain Areal Activity with the Entire Brain’s Activity"

_brainsci, 2022, doi:10.3390/brainsci13010006_

Round 1

Reviewer 1 Report

The author here established an objective and automatic method (shown as Eq.6 in the manuscript) for identification of functional correlated brain networks and characterization of entire brain's activity at large scale. The manuscript clearly stated the mathematical approach by introducing several crucial parameters (r, R, Rr, R(r), a, D(r)) for depicting brain activity correlation in different ways. Also, the author conclusion accurately summarized the major results and implications including: 

1. brain activity showed significant variation in terms of brain states and tested subjects (Fig 3); 

2. All FAUPAs with the same FC maps establish architechtural units of a functional network. 

Author Response

Thank you for reviewing this manuscript.

Reviewer 2 Report

General comment

The author proposes a statistical approach for the quantitative evaluation of  gradient-echo MRI  measurements of brain activities. The approach is presented using a fMRI BOLD data set obtained in a clinical study with nine healthy participants in resting state of brain activity and in task activated state. The author introduces entire brain BOLD measurements as reference value. The method is based on correlation analyses between time course and spatial distribution in so called functional areas of unitary  pooled activity (FAUPA)  and regions obtained functional coactivity (FC) maps of the entire brain.  The claim of  the author in the abstract is to ensure that the method is able to reflect the neuronal activity in the task specific areas in comparison to non-activated areas in other parts of the brain. In general, a discussion on restrictions of the method is not provided with the argument that the determination of the data employed was objective and  automatic (Introduction line 36; Discussion, lines 270/271). However, today also the requirements for accurate source localization  e.g. laminar fMRI are high. and a more complex view on the local vascular system in the focus of BOLD can be expected.

Major comments

-The reviewer is missing comparison and discussion on current methods for evaluation of spatial and time dependent fMRI based activity distribution and the advantages expected from the approach presented here.

-Four of the nine references are papers by the author himself and the other references were published 20-30 years ago. A broader literature study should allow an evaluation of the approach also in relation to other and also to more current work.

-Would be interesting also if the method was applied perhaps to a model system and in which signal intensities/thresholds could be verified  that the reliability of the measurements can be optimal.

-The claim of the author in the abstract that  BOLD reflects neuronal activity ignores one of the problems in fMRI. GE-BOLD signal is regarded today as biased towards ascending veins and large veins on the pial surface. For review see for instance Yang et al. Neuroscience and Biobehavioral Reviews 128 (2021) 467-478.

-Information/discussion  by the author how far a possible bias due to  such interferences can  be of importance in the evaluation approach presented would be desirable.

Reviewer 3 Report

In the manuscript entitled-‘The relationship of brain areal activity with the entire brain’s activity’, the author has proposed an excellent approach to quantify the entire brain’s activity quantitatively for a brain functional state in individual humans employing fMRI study. The work is interesting and based on the previous study of the author's lab. The manuscript is well-structured, scientifically sound, and easy to understand. Methods are sufficiently described and the conclusions are based on the critical discussion of the results. The manuscript is appropriate to be considered in my opinion. 

Author Response

(The authors gave the same response as above.)

Reviewer 4 Report

This paper “The relationship of brain areal activity with the entire brain’s activity” aimed to show a quantitative relationship between brain areal activity and the activity of the entire brain. The authors narrate that the temporal correlation coefficient r of the signal time courses of two areas quantifies the degree of co-activity between the two areas, and the spatial correlation coefficient R of their corresponding two FC maps quantifies the co-activity between these two maps. The authors also claimed that they found a modified sigmoid function quantified this R with r, revealing a relationship between the activity of brain areas and that of the entire brain. Their study demonstrated an approach to analyzing fMRI data to holistically characterize the entire brain’s activity quantitatively for any brain functional state in individual humans.

The topic is justified. The paper could be further improved if the following remarks are taken into consideration:

1.       The title of the study should be more elaborative, as I observe the suggested approach to analyzing fMRI data to holistically characterize the entire brain’s activity quantitatively in this article, but currently the title does not reflect it well.

2.       ABSTRACT: The text should include more details about the proposed methodology, numerical results achieved, and a comparison with other state-of-the-art methods (if established).

3.       Few grammatical mistakes were found in the whole draft of the article; the authors need to fix these.

4.       Introduction section lacks a proper introduction of the whole of the conducted research, background, justification of the research, and major contributions of the study. The contribution may be key fold in the introduction section.

5.       Add a recently conducted related to establishing the research problem.

6.       The study on only nine (9) controls could not ensure robustness, if possible, scan more controls to make them part of the study. Authors need to mention the volume of the dataset, explicitly.

7.       Image Pre-processing: what was the impact if functional images were not pre-processed?

8.       FAUPA determination seems ok.

9.       Did the authors verify, what is the relationship between the TC r and SC R, of abnormal (with any of the brain disordered, i.e., psychological, degenerative, inflammatory, Neoplastic, Cerebrovascular, etc.) subjects?

10.   The authors need to explicitly mention their baseline standards, to which the reported analysis is better radiologically.

11.   The motivation is not clear. Please specify the importance of the proposed solution.

12.   Discuss the limitations of the proposed method with their possible solutions in the future work section.

13.   The authors are required to review recent literature, the reference section contains too old stuff.
